# Model-based Bayesian inference of neural activity and connectivity from all-optical interrogation of a neural circuit

**Laurence Aitchison**
University of Cambridge
Cambridge, CB2 1PZ, UK
laurence.aitchison@gmail.com

**Lloyd Russell**
University College London
London, WC1E 6BT, UK
llerussell@gmail.com

**Adam Packer**
University College London
London, WC1E 6BT, UK
adampacker@gmail.com

**Jinyao Yan**
Janelia Research Campus
Ashburn, VA 20147
yanj11@janelia.hhmi.org

**Philippe Castonguay**
Janelia Research Campus
Ashburn, VA 20147
ph.castonguay@gmail.com

**Michael Häusser**
University College London
London, WC1E 6BT, UK
m.hausser@ucl.ac.uk

**Srinivas C. Turaga**
Janelia Research Campus
Ashburn, VA 20147
turagas@janelia.hhmi.org

## Abstract

Population activity measurement by calcium imaging can be combined with cellular resolution optogenetic activity perturbations to enable the mapping of neural connectivity *in vivo*. This requires accurate inference of perturbed and unperturbed neural activity from calcium imaging measurements, which are noisy and indirect, and can also be contaminated by photostimulation artifacts. We have developed a new fully Bayesian approach to jointly inferring spiking activity and neural connectivity from *in vivo* all-optical perturbation experiments. In contrast to standard approaches that perform spike inference and analysis in two separate maximum-likelihood phases, our joint model is able to propagate uncertainty in spike inference to the inference of connectivity and *vice versa*. We use the framework of variational autoencoders to model spiking activity using discrete latent variables, low-dimensional latent common input, and sparse spike-and-slab generalized linear coupling between neurons. Additionally, we model two properties of the optogenetic perturbation: off-target photostimulation and photostimulation transients. Using this model, we were able to fit models on 30 minutes of data in just 10 minutes. We performed an all-optical circuit mapping experiment in primary visual cortex of the awake mouse, and use our approach to predict neural connectivity between excitatory neurons in layer 2/3. Predicted connectivity is sparse and consistent with known correlations with stimulus tuning, spontaneous correlation and distance.

## 1 Introduction

Quantitative mapping of connectivity is an essential prerequisite for understanding the operation of neural circuits. Thus far, it has only been possible to perform neural circuit mapping by using electrophysiological [1, 2], or electron-microscopic [3, 4] techniques. In addition to being extremely

involved, these techniques are difficult or impossible to perform *in vivo*. But a new generation of all-optical techniques enable the simultaneous optical recording and perturbation of neural activity with cellular resolution *in vivo* [5]. In principle, cellular resolution perturbation experiments can enable circuit mapping *in vivo*, however several challenges exist.

First, while two-photon optogenetics can be used to drive spikes in neurons with cellular resolution, there can be variability in the number of spikes generated from trial to trial and from neuron to neuron. Second, there can be substantial off-target excitation of neurons whose dendrites might pass close to the targeted neurons. Third, there is a transient artifact from the laser pulse used for photostimulation which contaminates the activity imaging, preventing accurate estimates of changes in neural activity at the precise time of the perturbation, when accurate activity estimates are most useful. Fourth, the readout of activity in the stimulated neurons, and their downstream neighbors is a noisy flourescence measurement of the intracellular calcium concentration, which is itself an indirect measure of spiking activity. Fifth, the synaptic input from one neuron is rarely strong enough to generate action potentials on its own. Thus the optogenetic perturbation of single neurons is unlikely to generate changes in the suprathreshold activity of post-synaptic neurons which can be detected via calcium imaging on every trial.

Highly sensitive statistical tools are needed to infer neural connectivity in the face of these unique challenges posed by modern all-optical experimental technology. To solve this problem, we develop a global Bayesian inference strategy, jointly inferring a distribution over spikes and unknown connections, and thus allowing uncertainty in the spikes to influence the inferred connections and *vice versa*. In the past, such methods have not been used because they were computationally intractable, but they are becoming increasingly possible due to three recent advances: the development of GPU computing [6], modern automatic differentiation libraries such as Tensorflow [7], and recent developments in variational autoencoders, including the reparameterization trick [8, 9]. By combining these techniques, we are able to perform inference in a large-scale model of calcium imaging data, including spike inference, photostimulation, low-dimensional activity, and generalized linear synaptic connectivity.

## 1.1 Prior work

Bayesian models have been proposed to infer connectivity from purely observational neural datasets [10, 11], however such approaches do not recover connectivity in the common setting where the population neural activity is low-rank or driven by external unobserved inputs. Perturbations are essential to uncover connectivity in such scenarios, and a combination of electrophysiological readout and optogenetic perturbation has been used successfully [12, 13]. The analysis of such data is far simpler than our setting as electrophysiological measurements of the sub-threshold membrane potential of a post-synaptic neuron can enable highly accurate detection of strong and weak incoming connections. In contrast, we are concerned with the more challenging setting of noisy calcium imaging measurements of suprathreshold post-synaptic spiking activity. Further, we are the first to accurately model artifacts associated with 2-photon optogenetic photostimulation and simultaneous calcium imaging, while performing joint inference of spiking neural activity and sparse connectivity.

## 2 Methods

### 2.1 Variational Inference

We seek to perform Bayesian inference, i.e. to compute the posterior over latent variables, $z$, (e.g. weights, spikes) given data, $x$ (i.e. the fluorescence signal),

$$\mathrm{P}\left(z|x\right) = \frac{\mathrm{P}\left(x|z\right)\mathrm{P}\left(z\right)}{\mathrm{P}\left(x\right)}, \tag{1}$$

and, for model comparison, we would like to compute the model evidence,

$$\mathrm{P}\left(x\right) = \int dz\,\mathrm{P}\left(x|z\right)\mathrm{P}\left(z\right). \tag{2}$$

However, the computation of these quantities is intractable, and this intractability has hindered the application of Bayesian techniques to large-scale data analysis, such as calcium imaging. Variational

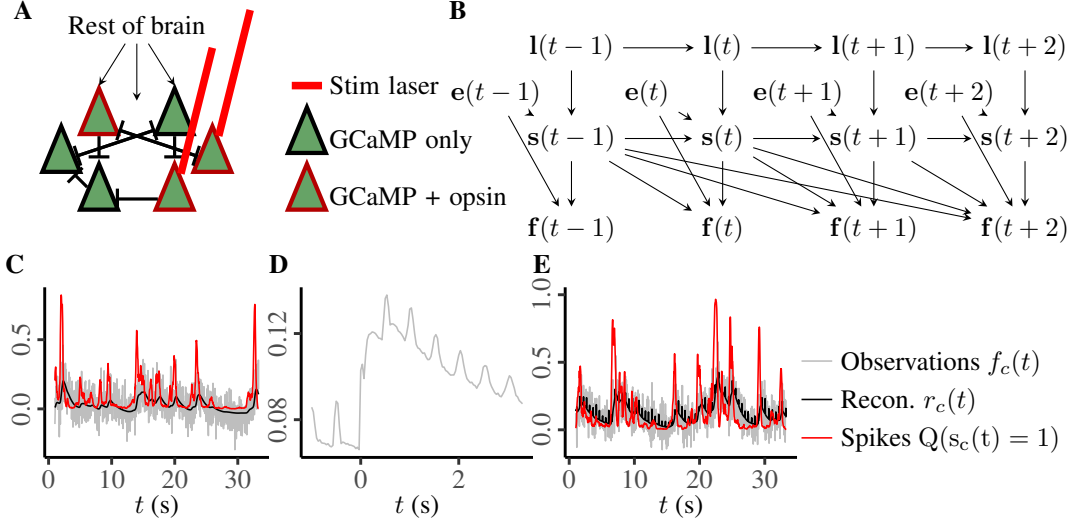

**Figure 1:** An overview of the data and generative model. **A.** A schematic diagram displaying the experimental protocol. All cells express a GCaMP calcium indicator, which fluoresces in response to spiking activity. A large subset of the excitatory cells also express channelrhodopsin, which, in combination with two-photon photostimulation, allows cellular resolution activity perturbations [5]. **B.** A simplified generative model, omitting unknown weights. The observed fluorescence signal, $\mathbf{f}$, depends on spikes, $\mathbf{s}$, at past times, and the external optogenetic perturbation, $\mathbf{e}$ (to account for the small photostimulation transient, which lasts only one or two frames). The spikes depend on previous spikes, external optogenetic stimulation, $\mathbf{e}$, and on a low-dimensional dynamical system, $\mathbf{l}$, representing the inputs coming from the rest of the brain. **C.** Results for spike inference based on spontaneous data. Gray gives the original (very noisy) fluorescence trace, black gives the reconstructed denoised fluorescence trace, based on inferred spikes, and red gives the inferred probability of spiking. **D.** Average fluorescence signal for cells that are directly perturbed (triggered on the perturbation). We see a large increase and slow decay in the fluorescence signal, driven by spiking activity. The small peaks at 0.5 s intervals are photostimulation transients. **E.** As in **C**, but for perturbed data. Note the small peaks in the reconstruction coming from the modelled photostimulation transients.

inference is one technique for circumventing this intractability [8, 9, 14], which, in combination with recent work in deep neural networks (DNNs), has proven extremely effective [8, 9]. In variational inference, we create a recognition model/approximate posterior, $Q\left(z|x\right)$, intended to approximate the posterior, $P\left(z|x\right)$ [14]. This recognition model allows us to write down the evidence lower bound objective (ELBO),

$$\log P\left(x\right) \geq \mathcal{L} = E_{Q(z|x)}\left[\log P\left(x, z\right) - \log Q\left(z|x\right)\right], \tag{3}$$

and optimizing this bound allows us to improve the recognition model, to the extent that, if $Q\left(z|x\right)$ is sufficiently flexible, the bound becomes tight and the recognition model will match the posterior, $Q\left(z|x\right) = P\left(z|x\right)$.

## 2.2 Our model

At the broadest possible level, our experimental system has known inputs, observed outputs, and unknown latent variables. The input is optogenetic stimulation of randomly selected cells (Fig. 1A; i.e. we target the cell with a laser, which usually causes it to spike), represented by a binary vector, $\mathbf{e}_t$, which is 1 if the cell is directly targeted, and 0 if it is not directly targeted. There are three unknown latent variables/parameters over which we infer an approximate posterior. First, there is a synaptic weight matrix, $\mathbf{W}^{\text{ss}}$, describing the underlying connectivity between cells. Second, there is a low-dimensional latent common input, $\mathbf{l}_t$, which represents input from other brain regions, and changes slowly over time (Fig. 1B). Third, there is a binary latent, $\mathbf{s}_t$, representing spiking activity, which depends on previous spiking activity through the synaptic weight matrix, optogenetic stimulation and the low-rank latent (Fig. 1B). Finally, we observe spiking activity indirectly through a flourescence signal, $\mathbf{f}_t$, which is in essence a noisy convolution of the underlying spikes. As such, the observations and latents can be written,

$$x = \mathbf{f},$$
$$z = \{\mathbf{l}, \mathbf{s}, \mathbf{W}^{\text{ss}}\},$$

respectively. Substituting these into the ELBO (Eq. 3), the full variational objective becomes,

$$\mathcal{L} = \mathrm{E}_{\mathrm{Q}(\mathbf{s},\mathbf{l},\mathbf{W}^{\mathrm{ss}}|\mathbf{f},\mathbf{e})} \left[ \log \mathrm{P} \left( \mathbf{f}, \mathbf{s}, \mathbf{l}, \mathbf{W}^{\mathrm{ss}}|\mathbf{e} \right) - \log \mathrm{Q} \left( \mathbf{s}, \mathbf{l}, \mathbf{W}^{\mathrm{ss}}|\mathbf{f}, \mathbf{e} \right) \right], \tag{4}$$

where we have additionally conditioned everything on the known inputs, $\mathbf{e}$.

## 2.3 Generative model

Neglecting initial states, we can factorize the generative model as

$$\mathrm{P} \left( \mathbf{f}, \mathbf{s}, \mathbf{l}, \mathbf{W}^{\mathrm{ss}}|\mathbf{e} \right) = \mathrm{P} \left( \mathbf{W}^{\mathrm{ss}} \right) \prod_t \mathrm{P} \left( \mathbf{l}_t|\mathbf{l}_{t-1} \right) \mathrm{P} \left( \mathbf{s}_t|\mathbf{s}_{t-1:0}, \mathbf{e}, \mathbf{l}_t, \mathbf{W}^{\mathrm{ss}} \right) \mathrm{P} \left( \mathbf{f}_t|\mathbf{s}_{t:0}, \mathbf{e}_t \right), \tag{5}$$

i.e., we first generate a synaptic weight matrix, $\mathbf{W}^{\mathrm{ss}}$, then we generate the latent low-rank states, $\mathbf{l}_t$ based on their values at the previous time-step, then we generate the spikes based on past spikes, the synaptic weights, optogenetic stimulation, $\mathbf{e}$, and the low-rank latents, and finally, we generate the flourescence signal based on past spiking and optogenetic stimulation. To generate synaptic weights, we assume a sparse prior, where there is some probability $p$ that the weight is generated from a zero-mean Gaussian, and there is probability $1 - p$ that the weight is zero,

$$\mathrm{P} \left( W_{ij}^{\mathrm{ss}} \right) = (1-p)\delta \left( W_{ij}^{\mathrm{ss}} \right) + p\mathcal{N} \left( W_{ij}^{\mathrm{ss}}, 0, \sigma^2 \right), \tag{6}$$

where $\delta$ is the Dirac delta, we set $p = 0.1$ based on prior information, and learn $\sigma^2$. To generate the low-rank latent states, we use a simple dynamical system,

$$\mathrm{P} \left( \mathbf{l}_t|\mathbf{l}_{t-1} \right) = \mathcal{N} \left( \mathbf{l}_t; \mathbf{W}^{\mathrm{ll}}\mathbf{l}_{t-1}, \mathbf{\Sigma}^{\mathrm{l}} \right). \tag{7}$$

where $\mathbf{W}^{\mathrm{ll}}$ is the dynamics matrix, and $\mathbf{\Sigma}^{\mathrm{l}}$ is a diagonal covariance matrix, representing independent Gaussian noise. To generate spikes, we use,

$$\mathrm{P} \left( \mathbf{s}_t|\mathbf{s}_{t-1:0}, \mathbf{e}, \mathbf{l}_t, \mathbf{W}^{\mathrm{ss}} \right) = \text{Bernoulli} \left( \mathbf{s}_t; \boldsymbol{\sigma} \left( \mathbf{u}_t \right) \right) \tag{8}$$

where $\boldsymbol{\sigma}$ is a vectorised sigmoid, $\sigma_i \left( \mathbf{x} \right) = 1/\left( 1 + e^{-x_i} \right)$, and the cell's inputs, $\mathbf{u}_t$, are given by,

$$\mathbf{u}_t = \mathbf{W}^{\mathrm{se}}\mathbf{e}_t + \mathbf{W}^{\mathrm{ss}} \sum_{t'=t-4}^{t-1} \kappa_{t-t'}^{\mathrm{s}}\mathbf{s}_{t'} + \mathbf{W}^{\mathrm{sl}}\mathbf{l}_t + \mathbf{b}^{\mathrm{s}}. \tag{9}$$

The first term represents the drive from optogenetic input, $\mathbf{e}_t$, (to reiterate, a binary vector representing whether a cell was directly targeted on this timestep), coupled by weights, $\mathbf{W}^{\mathrm{se}}$, representing the degree to which cells surrounding the targeted cell also respond to the optogenetic stimulation. Note that $\mathbf{W}^{\mathrm{se}}$ is structured (i.e. written down in terms of other parameters), and we discuss this structure later. The second term represents synaptic connectivity: how spikes at previous timesteps, $\mathbf{s}_{t'}$ might influence spiking at this timestep, via a rapidly-decaying temporal kernel, $\kappa^{\mathrm{s}}$, and a synaptic weight matrix $\mathbf{W}^{\mathrm{ss}}$. The third term represents the input from other brain-regions by allowing the low-dimensional latents, $\mathbf{l}_t$, to influence spiking activity according to a weight matrix, $\mathbf{W}^{\mathrm{sl}}$. Finally, to generate the observed flourescence signal from the spiking activity, we use,

$$\mathrm{P} \left( \mathbf{f}_t \right) = \mathcal{N} \left( \mathbf{f}_t; \mathbf{r}_t, \mathbf{\Sigma}^{\mathrm{f}} \right), \tag{10}$$

where $\mathbf{\Sigma}^{\mathrm{f}}$ is a learned, diagonal covariance matrix, representing independent noise in the flourescence observations. For computational tractability, the mean flourescence signal, or "reconstruction", is simply a convolution of the spikes,

$$\mathbf{r}_t = \mathbf{A} \sum_{t'=0}^{t} \boldsymbol{\kappa}_{t-t'} \odot \mathbf{s}_{t'} + \mathbf{b}^{\mathrm{r}} + \mathbf{W}^{\mathrm{re}}\mathbf{e}_t, \tag{11}$$

where $\odot$ represents an entrywise, or Hadamard, product. This expression takes a binary vector representing spiking activity, $\mathbf{s}_{t'}$, convolves it with a temporal kernel, $\boldsymbol{\kappa}$, representing temporal dynamics of flourescence responses, then scales it with the diagonal matrix, $\mathbf{A}$, and adds a bias, $\mathbf{b}^{\mathrm{r}}$. The last term models an artifact in which optogenetic photostimulation, represented by a binary vector $\mathbf{e}_t$ describing whether a cell was directly targeted by the stimulation laser on that timestep, directly affects the imaging system according to a weight matrix $\mathbf{W}^{\mathrm{re}}$. The temporal kernel, $\kappa_{c,t-t'}$ is a sum of two exponentials unique to each cell,

$$\kappa_{c,t} = e^{-t/\tau_c^{\mathrm{decay}}} - e^{-t/\tau_c^{\mathrm{rise}}}, \tag{12}$$

as is typical in e.g. [15].

## 2.4 Recognition model

The recognition model factorises similarly,

$$Q\left(\mathbf{s}, \mathbf{l}, \mathbf{W}^{ss} | \mathbf{f}, \mathbf{e}\right) = Q\left(\mathbf{W}^{ss}\right) Q\left(\mathbf{s} | \mathbf{f}, \mathbf{e}\right) Q\left(\mathbf{l} | \mathbf{f}\right). \tag{13}$$

To approximate the posterior over weights we use,

$$Q\left(W_{ij}^{ss}\right) = (1 - p_{ij})\delta\left(W_{ij}^{ss}\right) + p_{ij}\mathcal{N}\left(W_{ij}^{ss}, \mu_{ij}, \sigma_{ij}^2\right). \tag{14}$$

where $p_{ij}$ is the inferred probability that the weight is non-zero, and $\mu_{ij}$ and $\sigma_{ij}^2$ are the mean and variance of the inferred distribution over the weight, given that it is non-zero. As a recognition model for spikes, we use a multi-layer perceptron to map from the flourescence signal back to an inferred probability of spiking,

$$Q\left(\mathbf{s}(t) | \mathbf{v}(t)\right) = \text{Bernoulli}\left(\mathbf{s}(t); \boldsymbol{\sigma}\left(\mathbf{v}(t)\right)\right), \tag{15}$$

where $\mathbf{v}(t)$ depends on the fluorescence trace, and the optogenetic input,

$$\mathbf{v}(t) = \text{MLP}^s\left(\mathbf{f}(t - T : t + T)\right) + \mathbf{D}^e \mathbf{W}^{se} \mathbf{e}(t) + \mathbf{b}^s. \tag{16}$$

Here, $\mathbf{D}^e$ is a diagonal matrix scaling the external input, and $\text{MLP}\left(\mathbf{f}(t - T : t + T)\right)$ is a neural network that, for each cell, takes a window of the fluorescence trace from time $t - T$ to $t + T$, (for us, $T = 100$ frames, or about 3 seconds) linearly maps this window onto 20 features, then maps those 20 features through 2 standard neural-network layers with 20 units and Elu non-linearities [16], and finally linearly maps to a single value. To generate the low-rank latents, we use the same MLP, but allow for a different final linear mapping from 20 features to a single output,

$$Q\left(\mathbf{l}(t) | \mathbf{f}\right) = \mathcal{N}\left(\mathbf{l}(t); \mathbf{W}^{fl} \text{MLP}^l\left(\mathbf{f}(t - T : t + T)\right), \boldsymbol{\Gamma}^l\right). \tag{17}$$

Here, we use a fixed diagonal covariance, $\boldsymbol{\Gamma}^l$, and we use $\mathbf{W}^{fl}$ to reduce the dimensionality of the MLP output to the number of latents.

## 2.5 Gradient-based optimization of generative and recognition model parameters

We used the automatic differentiation routines embedded within TensorFlow to differentiate the ELBO with respect to the parameters of both the generative and recognition models,

$$\mathcal{L} = \mathcal{L}\left(\sigma, \mathbf{W}^{ll}, \boldsymbol{\Sigma}^l, \mathbf{W}^{sl}, \mathbf{b}^s, \boldsymbol{\Sigma}^f, \tau_c^{\text{decay}}, \tau_c^{\text{rise}}, \mathbf{b}^r, \mathbf{W}^{re}, p_{ij}, \mu_{ij}, \sigma_{ij}^2, \mathbf{D}^e, \mathbf{W}^{fl}, \text{MLP}, \text{resp}_i, \sigma_k\right), \tag{18}$$

where the final two variables are defined later. We then used Adam [17] to perform the optimization. Instead of using minibatches consisting of multiple short time-windows, we used a single, relatively large time-window (of 1000 frames, or around 30 s, which minimized any edge-effects at the start or end of the time-window.

# 3 Results

## 3.1 All-optical circuit mapping experimental protocol

We used a virus to express GCaMP6s pan-neuronally in layer 2/3 of mouse primary visual cortex (V1), and co-expressed C1V1 in excitatory neurons of the same layer. The mouse was awake, headfixed and on a treadmill. As in [5], we used a spatial light modulator to target 2-photon excitation of the C1V1 opsin in a subset of neurons, while simultaneously imaging neural activity in the local circuit by 2-photon calcium imaging of GCaMP6s. With this setup, we designed an experimental protocol to facilitate discovery of a large portion of the connections within a calcium-imaging field of view. In particular, twice every second we selected five cells at random, stimulated them, observed the activity in the rest of the network, and used this information to infer whether the stimulated cells projected to any of the other cells in the network (Fig. 1A). The optogenetic perturbation experiment consisted of 7200 trials and lasted one hour. We also mapped the orientation and direction tuning properties of the imaged neurons, and separately recorded spontaneous neural activity for 40 minutes. Our model was able to infer spikes in spontaneous data (Fig. 1C), and in photostimulation data, was able to both infer spikes and account for photostimulation transients (Fig. 1DE).

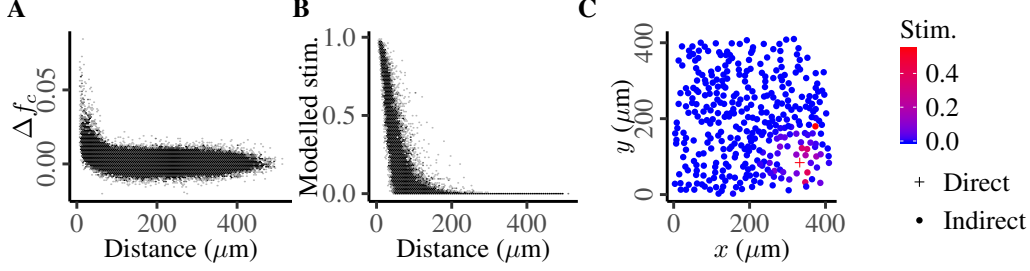

**Figure 2:** Modeling off-target photostimulation, in which stimulating at one location activates surrounding cells. **A.** The change in average fluorescence based on 500 ms just before and just after stimulation ($\Delta f_c$) for photostimulation of a target at a specified distance [5]. **B.** The modelled distance-dependent activation induced by photostimulation. The spatial extent of modelled off-target stimulation is broadly consistent with the raw-data in **A**. Note that as each cell has a different spatial absorption profile and responsiveness, modelled stimulation is not a simple function of distance from the target cell. **C.** Modelled off-target photostimulation resulting from stimulation of an example cell.

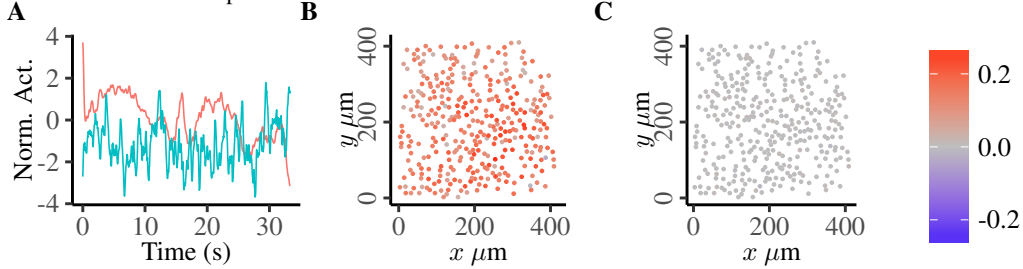

**Figure 3:** Inferred low-rank latent activity. **A.** Time course of $l_t$ for perturbed data. The different lines correspond to different modes. **B.** The projection weights from the first latent onto cells, where cells are plotted according to their locations on the imaging plane. **C.** As **B** but for the second latent. Note that all projection weights are very close to 0, so the points are all gray.

## 3.2 Inferring the extent of off-target photostimulation

Since photostimulation may also directly excite off-target neurons, we explicitly modelled this process (Fig. 2A). We used a sum of five Gaussians with different scales, $\sigma_k$, to flexibly model distance-dependent stimulation,

$$\mathbf{W}^{\text{se}}_{ij} = \text{resp}_i \sum_{k=1}^{5} \exp\left[d_i^2(\mathbf{x}_j)/\left(2\sigma_k^2\right)\right],\tag{19}$$

where $\mathbf{x}_j$ describes the $x, y$ position of the "target" cell $j$, and each cell receiving off-target stimulation has its own degree of responsiveness, $\text{resp}_i$, and a metric, $d_i(x_j, y_j)$, describing that cell's response to light stimulation in different spatial locations. The metric allows for stimulation to take on an elliptic pattern (given by $\mathbf{P}_i$'s), and have a shifted center (given by $\hat{\mathbf{x}}_i$),

$$d_i^2(\mathbf{x}_j) = (\mathbf{x}_j - \hat{\mathbf{x}}_i)^T \mathbf{P}_i (\mathbf{x}_j - \hat{\mathbf{x}}_i)\tag{20}$$

After inference, this model gives a similar spatial distribution of perturbation-triggered activity (Fig. 2B). Furthermore, it should be noted that because each cell has its own responsiveness and spatial light absorption profile, if we stimulate in one location, a cell's responsiveness is not a simple function of distance (Fig. 2BC). Finally, we allow small modifications around this strict spatial profile using a dense weight matrix.

## 3.3 Joint inference of latent common inputs

Our model was able to jointly infer neural activity, latent common inputs (Fig. 3A) and sparse synaptic connectivity. As expected, we found one critical latent variable describing overall activation of all cells (Fig. 3B) [18], and a second, far less important latent (Fig. 3C). Given the considerable difference in magnitude between the impact of these two latents on the system, we can infer that only one latent variable is required to describe the system effectively. However, further work is needed to implement flexible yet interpretable low-rank latent variables in this system.

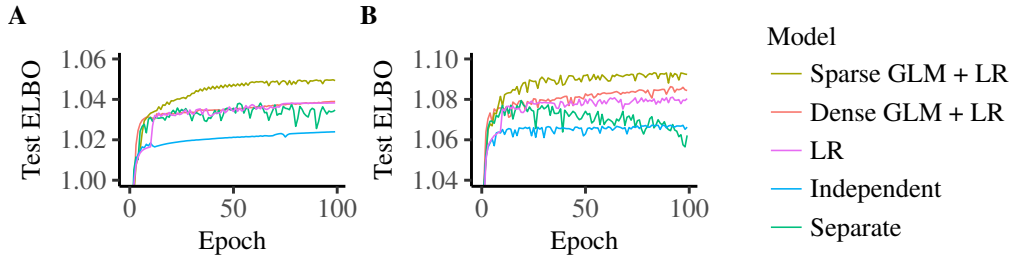

**Figure 4:** Performance of various models for spontaneous (**A**) and perturbed (**B**) data. We consider "Sparse GLM + LR" (the full model), "Dense GLM + LR" (the full model, but with with dense GLM weights), "LR" (a model with no GLM, only the low-rank component), "Independent" (a model with no higher-level structure) and finally "Separate" (the spikes are extracted using the independent model, then the full model is fitted to those spikes).

### 3.4    The model recovers known properties of biological activity

The ELBO forms only a lower bound on the model evidence, so it is possible for models to appear better/worse simply because of changes in the tightness of the bound. As such, it is important to check that the learned model recovers known properties of biological connectivity. We thus compared a group of models, including the full model, a model with dense (as opposed to the usual sparse) synaptic connectivity, a model with only low-rank latents, and a simple model with no higher-level structure, for both spontaneous (Fig. 4A) and perturbed (Fig. 4B) data. We found that the sparse GLM offered a dramatic improvement over the dense GLM, which in turn offered little benefit over a model with only low-rank activity. (Note the reported values are ELBO per cell per timestep, so must be multiplied by 348 cells and around 100,000 time-steps to obtain the raw-ELBO values, which are then highly significant). Thus, the ELBO is able to recover features of real biological connectivity (biological connectivity is also sparse [1, 2]).

### 3.5    Joint inference is better than a "pipeline"

Furthermore, we compared our joint approach, where we jointly infer spikes, low-rank activity, and weights, to a more standard "pipeline" in which one infer spikes using a simple Bayesian model lacking low-rank activity and GLM connectivity, then infer the low-rank activity and weights based on those spikes, similar to [11]. We found that performing inference jointly — allowing information about low-rank activity, GLM connectivity and external stimulation to influence spike inferences greatly improved the quality of our inferences for both spontaneous (Fig. 4A) and perturbed data (Fig. 4B). This improvement is entirely expected within the framework of variational inference, as the "pipeline" has two objectives, one for spike extraction, and another for the high-level generative model, and without the single, unified objective, it is even possible for the ELBO to decrease with more training (Fig. 4B).

### 3.6    The inferred sparse weights are consistent with known properties of neural circuits

Next, we plotted the synaptic "GLM" weights for spontaneous (Fig. 5A–D) and perturbed (Fig. 5E–H) data. These weights are negatively correlated with distance ($p < 0.0001$; Fig. 5BF) suggesting that short-range connections are predominantly excitatory (though this may be confounded by cells overlapping, such that activity in one cell is recorded as activity in a different cell). The short range excitatory connections can be seen as the diagonal red bands in Fig. 5AE as the neurons are roughly sorted by proximity, with the first 248 being perturbed, and the remainder never being perturbed. The weights are strongly correlated with spontaneous correlation ($p < 0.0001$; Fig. 5CG), as measured using raw fluorescence traces; a result which is expected, given that the model should use these weights to account for some aspects of the spontaneous correlation. Finally, the weights are positively correlated with signal correlation ($p < 0.0001$; Fig. 5DH), as measured using 8 drifting gratings, a finding that is consistent with previous results [1, 2].

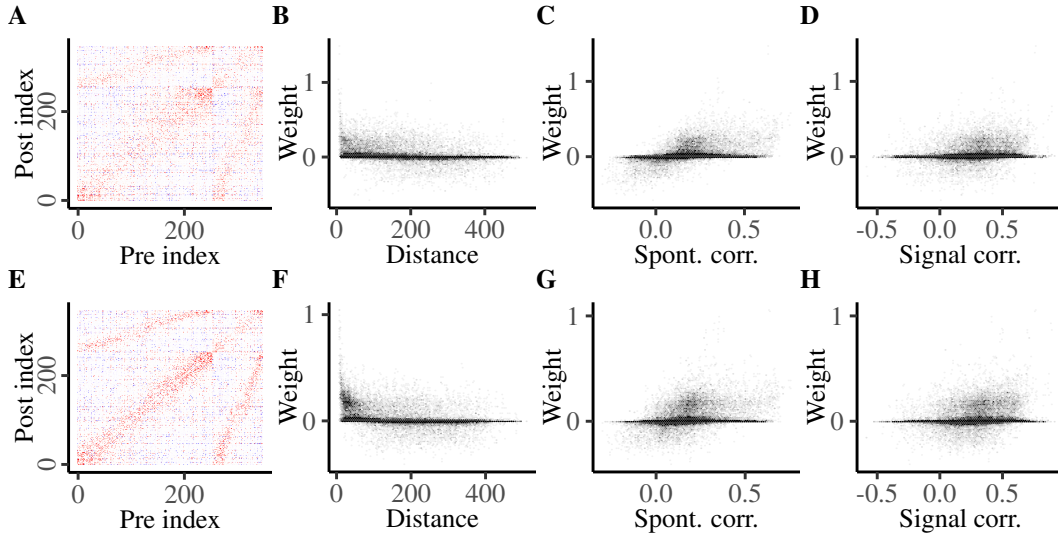

**Figure 5:** Inferred connection weights. **A.** Weight matrix inferred from spontaneous data (in particular, the expected value of the weight, under the recognition model, with red representing positive connectivity, and blue representing negative connectivity), plotted against distance (**B**), spontaneous correlation (**C**), and signal correlation (**D**). **E–H.** As **A–D** for perturbed data.

## 3.7 Perturbed data supports stronger inferences than spontaneous data

Consistent with our expectations, we found that perturbations considerably increased the number of discovered connections. Our spike-and-slab posterior over weights can be interpreted to yield an estimated confidence probability that a given connection exists. We can use this probability to estimate the number of highly confident connections. In particular, we were able to find 50% more connections in the perturbed dataset than the spontaneous dataset, with a greater than 0.95 probability (1940 vs 1204); twice times as many highly confident connections with probability 0.99 or higher (1107 vs 535); and five times as many with the probability 0.999 or higher (527 vs 101). These results highlight the importance of perturbations to uncovering connections which would otherwise have been missed when analyzing purely observational datasets.

## 3.8 Simulated data

Using the above methods, it is difficult to assess the effectiveness of the model because we do not have ground truth. While the ideal approach would be to obtain ground-truth data experimentally, this is very difficult in practice. An alternative approach is thus to simulate data from the generative model, in which case the ground-truth weights are simply those used to perform the initial simulation. To perform a quantitative comparison, we used the correlation between a binary variable representing whether the true weights were greater than 0.1 (because it is extremely difficult to distinguish between zero, and very small but non-zero weights, and), and the inferred probability of the weight being greater than 0.1, based on a combination of the inferences over the discrete and continuous component. We chose a threshold of 0.1 because it was relatively small in comparison with the standard-deviation for the non-zero weights of around 0.4. We started by trying to replicate our experiments as closely as possible (Fig. 6), i.e. we inferred all the parameters, noise-levels, timescales, priors on weights etc. based on real data, and resampled the weight matrix based on the inferred prior over weights. We then considered repeating the same stimulation pattern 50 times (frozen), as against using 50 times more entirely random simulated data (unfrozen), and found that, as expected, using random stimulation patterns is more effective. As computational constraints prevent us from increasing the data further, we considered reducing the noise by a factor of 40 (low-noise), and then additionally reduced the timescales of the calcium transients by a factor of 10 (fast decay) which improved the correlation to 0.85.

These results indicate the model is functioning correctly, but raise issues for future work. In particular, the considerable improvement achieved by reducing the timescales indicates that careful modeling of the calcium transient is essential, and that faster calcium indicators have the potential to dramatically improve the ultimate accuracy of weight inferences.

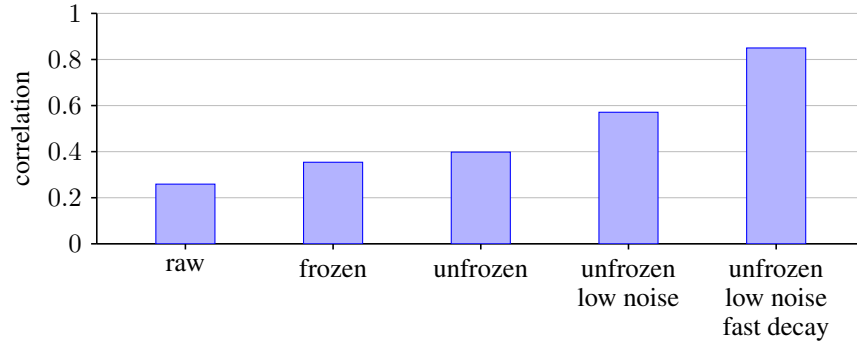

**Figure 6:** Effectiveness of various variants of the model at finding the underlying ground-truth weights. The correlation compares a binary variable reporting whether the ground-truth weight is above or below 0.1 with a continuous measure reporting the inferred probability of the weight being larger than 0.1. The first condition, raw, uses simulated data that matches the real data as closely as possible including the same length of photostimulated and spontaneous data as we obtained, and matching the parameters such as the noise level to those used in data. The frozen/unfrozen conditions represent using 50 times more data, where, for "frozen" condition, we repeat the same optogenetic stimulation 50 times, and for the "unfrozen" condition we always use fresh, randomly chosen stimulation patterns. The final pair of conditions are photo stimulated data, with 50 times more unfrozen data. For the "low noise" condition we reduce the noise level by a factor of 40, and for the "fast decay" condition, we additionally reduce the calcium decay timeconstants by a factor of 10.

## 4 Discussion

We applied modern variational autoencoder and GPU computing techniques to create a fully Bayesian model of calcium imaging and perturbation data. This model simultaneously and efficiently extracted Bayesian approximate posteriors over spikes, the extent of two optogenetic perturbation artifacts, low-rank activity, and sparse synaptic (GLM) weights. This is the first model designed for perturbation data, and we are not aware of any other model which is able to extract posteriors over such a wide range of parameters with such efficiency.

Our inferred weights are consistent with studies using electrophysiological means to measure connectivity in mouse V1 [1, 2]. Further, model selection gives biologically expected results, identifying sparseness, suggesting that these models are identifying biologically relevant structure in the data. However, simply identifying broad properties such as sparseness does not imply that our inferences about individual weights are correct: for this, we need validation using complementary experimental approaches.

Finally, we have shown that recent developments in variational autoencoders make it possible to perform inference in "ideal" models: large-scale models describing noisy data-generating processes and complex biological phenomena simultaneously.

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
