[Supplementary Material · supplementary.pdf]

## Supplementary Information: Sampling-free estimation of expectations with discrete latents

The model involves many discrete random variables, including the spikes and the variables representing the presence or absence of a synaptic connection. Once discrete variables are present, it is no longer possible to use the "reparameterization trick", meaning that we can no longer optimize the ELBO using simple stochastic gradient ascent, potentially dramatically slowing down learning [19, 20, 21]. However, we developed strategies that enabled us to compute expectations over these discrete variables without sampling, so we were able to retain the simplicity and speed of simple stochastic gradient descent. In particular, differentiating with respect to two terms was problematic: $\mathrm{E_Q}\left[\log \mathrm{P}\left(\mathbf{f}|\mathbf{s}\right)\right]$, and $\mathrm{E_Q}\left[\log \mathrm{P}\left(\mathbf{s}(t)|\mathbf{u}(t)\right)\right]$. Differentiating the first of these,

$$\mathrm{E_Q}\left[\log \mathrm{P}\left(f_{tc}|\mathbf{s}\right)\right] = -\mathrm{E_Q}\left[\left(f_{tc} - r_{tc}\right)^2\right] / \left(2\sigma_c^2\right) + \text{const} \tag{21}$$

using sample-based estimates is not possible, as the reconstruction depends on discrete spikes, $\mathbf{s}$. Instead, we can compute the expectation over spikes directly and exactly by noting that the log-likelihood depends only on the mean and variance of $r_{tc}$,

$$\mathrm{E_Q}\left[\log \mathrm{P}\left(f_{tc}|\mathbf{s}\right)\right] = -\mathrm{E_Q}\left[\left(f_{tc} - \mathrm{E_{Q(\mathbf{s}|\mathbf{f})}}\left[r_{tc}\right]\right)^2 + \mathrm{Var_{Q(\mathbf{s})}}\left[r_{tc}\right]\right] / \left(2\sigma_c^2\right) + \text{const} \tag{22}$$

and if (as in our case), the mapping from spikes to the reconstruction is linear, and the recognition model is factorised, we can compute these quantities simply by summing the mean and variance of spikes themselves,

$$\mathrm{E}\left[r_{tc}(\mathbf{s})\right] = \sum_{t'=0}^{t} \kappa_c(t-t')\mathrm{E_Q}\left[s_{t'c}\right], \qquad \mathrm{Var}\left[r_{tc}(\mathbf{s})\right] = \sum_{t'=0}^{t} \kappa_c^2(t-t')\mathrm{Var_Q}\left[s_{t'c}\right]. \tag{23}$$

Second, differentiating $\mathrm{E_Q}\left[\log \mathrm{P}\left(\mathbf{s}(t)|\mathbf{u}(t)\right)\right]$ using sample-based estimates is again not possible as there are several discrete random variables that form part of the input, $\mathbf{u}(t)$, including spikes from previous timesteps, and the binary variables representing the presence or absence of sparse GLM connections (Eq. 9). We therefore approximated the complex, discrete distribution over $\mathbf{u}(t)$ as Gaussian, with matched mean and variance (note that once we make this approximation, we are no longer guaranteed to bound the model evidence, so we are careful to use the sampled estimate when evaluating our model). We then need to compute,

$$
\begin{aligned}
\mathrm{E}_{\mathrm{Q}(u_c(t))} & \left[\log \mathrm{P}\left(\mathbf{s}(t)|\mathbf{u}(t)\right)\right] \\
&= \sum_c \mathrm{E}_{\mathrm{Q}(u_c(t))}\left[s_c(t)\log\sigma\left(u_c(t)\right) + (1-s_c(t))\log\sigma\left(-u_c(t)\right)\right], \\
&= -\sum_c \mathrm{E}_{\mathrm{Q}(u_c(t))}\left[s_c(t)\mathrm{softplus}\left(-u_c(t)\right) + (1-s_c(t))\,\mathrm{softplus}\left(u_c(t)\right)\right],
\end{aligned} \tag{24}
$$

where $\mathrm{softplus}\left(x\right) = \log\left(1+e^x\right)$. As the expectation of a softplus under Gaussian distributed input cannot (to our knowledge) be computed analytically, we used an upper-bound to the softplus (and hence a lower-bound to the ELBO) formed by a sum of ReLU's (Fig. S1A). While the assumption of Gaussianity means that this estimate is not guaranteed to be a lower-bound on either the model evidence or the ELBO, in practice, the resulting approximation does appear to be such a bound, in both spontaneous (Fig. S1B) and perturbed (Fib. S1C) data.

**A**

**B**

**C**

— $f(x) = \text{softplus}(x)$

— $f(x) = c + \sum_i g_i \text{relu}(x - \theta_i)$

— Sampled unbiased estimate

— Deterministic bound

**Figure S1:** Integrating over discrete, stochastic inputs to cells. **A.** We put an upper bound on the softplus function (blue) using a sum-of-ReLU's (red). In practice, we use 14 piecewise linear components at learned locations, allowing us to tightly bound the softplus. This approximation is critical, because we can compute the expectation of the sum-of-ReLU bound under Gaussian inputs. **BC.** The ELBO for spontaneous (**B**) and perturbed (**C**) data, computed using using the usual sample-based unbiased estimate and our deterministic approximation. In practice, our approximation appears to form a tight lower bound on the sampled estimate, as required.