[Reviews · NeurIPS 2017]

Reviewer 1



This papers proposes an inference method of (biological) neural connectivity from fluorescence (calcium ) traces. The model includes the spiking model (GLM+low-rank factor) with an external input (optical stimulation) and a fluorescence model. The inference methods is based on variational Bayes, where the approximate posterior is modeled using a neural network. Novelty and originality: The methods in this paper are adequately novel and original, nicely combining various elements from previous work. Technical issues: My main problem with this paper is that I can't really be sure that the proposed method is actually working well. It is very good that the authors tested their method on real data, but since there is no ground truth, I it is hard to estimate the quality of the inferred weights (see footnote (1) below). Given that there is no ground truth, what would really help to convince the reader that everything is working as it should, is some basic check on simulated data to show that the inferred weights are reasonably accurate. Also, it is hard to interpret Fig 5 due to its scale: how can the reader know that an improvement from 1.02 to 1.05 in the ELBO is significant? Clarity: The methods part seems a bit dense, with various details not completely clear from the text. I would suggest adding more complete details in the supplementary, such as the complete objective being optimized. This would also make it easier for the reader to understand the context of section 2.5. Also, from section 2.1 (e.g., lines 69-70) I thought that the weights are also treated as latent variables, and so have a recognition model. But since no recognition model is specified for the weights, so I guess I was wrong. Minor issues: 1) In Figure 2A, the approximate "sum of ReLU" softplus, we have c > 0, i.e. it to does not go to zero for low firing rates. Maybe using an exponential tail in that section would improve the approximation (at is still integrates with a Gaussian, and can upper bound a logistic tail). 2) For Gaussians, the notation is supposed to be N(mu,sigma^2), not N(mu,sigma). 3) line 192: Fig 6B -> Fig 6E. 4) Figure 6 third line in caption: remove "A." 5) The index 'c' is for "cell"? This should be made clearer. Footnote: (1) The authors argue that their method works since the properties of the detected weights match the expected biological statistics (Fig. 6), but this does not seem convincing. Specifically, It is a good sanity check, but these results only indicate that the method does not fail completely: it is not hard to think of cases in which we get the same properties, yet the estimate weights are highly inaccurate. For example, the observed distance dependence (Fig 6B) could be related to partial mixing of fluorescence signal of nearby neurons. With stimulation this could also be related to the "area affect" of the stimulation (Fig 6F), though this is partially compensated by the method. Similarly, the correlation of weight strength to strong Spont. and Signal. correlations might be the result that all of these factors typically increase with the firing rate of the neuron. %%% After authors' feedback %%% The new results and clarifications address my concerns. I therefore have modified my score. I would like to see the limit that the correlation -> 1 on the synthetic data set in the revised paper, to make sure the method is consistent.

Reviewer 2



This paper applies variational autoencoders (with all the latest tricks) to the problem of estimating connectivity between neurons from Calcium imaging data with and without optogenetic perturbations. This is an excellent paper with no major flaws as far as I can tell. My only issue with the paper is that it confounds sparse connectivity with biologically plausible connectivity in a few places (eg. Line 232). Just because the connectivity is sparse doesn’t mean that it’s anatomically accurate. The correlations in Fig 6 give some indication that the connectivity is realistic, but my guess is that something like graphical lasso would also show these patterns. What would be more compelling is some results on reconstruction accuracy (with leave-neuron-out cross-validation). This could help clarify the practical extent of the model improvements, but, otherwise, this is a great paper.

Reviewer 3



Summary This paper tackles the problem of inferring the functional connectivity of the brain, combining the recordings of neural activity by calcium imaging with optogenetic activity perturbations at a cellular level. The authors propose a Bayesian generative approach to jointly model the spiking neural activity, the optogenetic perturbations, the calcium imaging measurement process, the synaptic connectivity and a low-rank manifold representing the activity in the rest of the brain. They resort to stochastic variational inference to approximate the posterior using sampling and the reparametrization trick to compute expectations when they cannot be computed in closed form. Finally, they perform several experiments showing the relevance of the model. Qualitative Assessment The paper is well written and it addresses a problem relevant in the neuroscience community. The introduction successes at motivating the problem of unraveling the functional connectivity of the brain based of optogenetic perturbations and calcium imaging recordings, and describe the key elements that a sensible model should tackle. In my opinion, the main novelty of the paper is the well-thought Bayesian generative model that they proposed. Regarding the inference approach, the authors use standard stochastic variational inference. My main concern is regarding the experiments that the authors propose to validate the model. In particular, in Figure 5 they use the ELBO to compare the different proposed models. The ELBO is just a lower bound of the evidence of the data given the model. However, when comparing different models with different complexity (different number of parameters/hyper-parameters) there is no guarantees that the tightness of the bound is similar to all of them, making the comparison to totally fair. %%% After authors' feedback %%% In light of the clarifications provided by the authors, I have modified my score. In the revised version I would like to see the experiments with synthetic data as well as a more in depth discussion of the consequences of using the ELBO for model selection.